# Comparison of Structural and Short Variants Detected by Linked-Read and Whole-Exome Sequencing in Multiple Myeloma

**DOI:** 10.3390/cancers13061212

**Published:** 2021-03-10

**Authors:** Ashwini Kumar, Sadiksha Adhikari, Matti Kankainen, Caroline A. Heckman

**Affiliations:** 1Institute for Molecular Medicine Finland-FIMM, HiLIFE-Helsinki Institute of Life Science, iCAN Digital Cancer Medicine Flagship, University of Helsinki, Tukholmankatu 8, 00290 Helsinki, Finland; ashwini.kumar@helsinki.fi (A.K.); sadiksha.adhikari@helsinki.fi (S.A.); 2iCAN Digital Precision Cancer Medicine, University of Helsinki, 00014 Helsinki, Finland; matti.kankainen@helsinki.fi; 3Medical and Clinical Genetics, University of Helsinki, Helsinki University Hospital, 00029 Helsinki, Finland; 4Translational Immunology Research Program and Department of Clinical Chemistry, University of Helsinki, 00290 Helsinki, Finland; 5Hematology Research Unit Helsinki, Department of Hematology, Helsinki University Hospital Comprehensive Cancer Center, 00290 Helsinki, Finland

**Keywords:** genomics, NGS, linked-read sequencing, whole-exome sequencing, RNA sequencing, structural variants, short variants, FISH, multiple myeloma

## Abstract

**Simple Summary:**

The wide variety of next-generation sequencing technologies requires thorough evaluation and understanding of their advantages and shortcomings of these different approaches prior to their implementation in a precision medicine setting. Here, we compared the performance of two DNA sequencing methods, whole-exome and linked-read exome sequencing, to detect large structural variants (SVs) and short variants in eight multiple myeloma (MM) patient cases. For three patient cases, matched tumor-normal samples were sequenced with both methods to compare somatic SVs and short variants. The methods’ clinical relevance was also evaluated, and their sensitivity and specificity to detect MM-specific cytogenetic alterations and other short variants were measured. Thus, this study systematically demonstrates and evaluates the performance of whole-exome and linked-read exome sequencing technologies for detecting genetic alterations to aid in selecting the optimal method for clinical application.

**Abstract:**

Linked-read sequencing was developed to aid the detection of large structural variants (SVs) from short-read sequencing efforts. We performed a systematic evaluation to determine if linked-read exome sequencing provides more comprehensive and clinically relevant information than whole-exome sequencing (WES) when applied to the same set of multiple myeloma patient samples. We report that linked-read sequencing detected a higher number of SVs (n = 18,455) than WES (n = 4065). However, linked-read predictions were dominated by inversions (92.4%), leading to poor detection of other types of SVs. In contrast, WES detected 56.3% deletions, 32.6% insertions, 6.7% translocations, 3.3% duplications and 1.2% inversions. Surprisingly, the quantitative performance assessment suggested a higher performance for WES (AUC = 0.791) compared to linked-read sequencing (AUC = 0.766) for detecting clinically validated cytogenetic alterations. We also found that linked-read sequencing detected more short variants (n = 704) compared to WES (n = 109). WES detected somatic mutations in all MM-related genes while linked-read sequencing failed to detect certain mutations. The comparison of somatic mutations detected using linked-read, WES and RNA-seq revealed that WES and RNA-seq detected more mutations than linked-read sequencing. These data indicate that WES outperforms and is more efficient than linked-read sequencing for detecting clinically relevant SVs and MM-specific short variants.

## 1. Introduction

Genetic alterations are broadly categorized into short variants, like single nucleotide substitutions and small insertions and deletions, and larger structural variants (SVs). Aberrations longer than 50 bp are defined as SVs [1] and can occur in single or multiple chromosomes as pairs of breakpoints resulting in duplications, deletions, insertions, inversions and translocations among others in the genome [2,3]. Identifying and understanding SVs and short variants are important for cancer diagnosis and prognosis and have been employed to study disease initiation, progression and tumor evolution for multiple cancer types [3,4,5,6,7]. Extensive genome sequencing studies have revealed the landscapes of short variants and SVs in cancers [3]. However, detecting SVs with more cost-effective and popular exome targeting sequencing methods has been challenging. Although technological advances have made the detection of SVs with the help of exome sequencing more feasible, the proportion of SVs detected in the human genome has remained underrepresented in cancer genome studies [8,9,10,11].

Short paired-end read-based whole-exome sequencing (WES) approaches have successfully been used to identify many cancer-causing variants [12,13]. Although WES has been mostly used for short variant detection, the method has also been employed to identify SVs. For example, computational tools such as SvABA, BreakDancer, Manta and DELLY have been developed to extract SV information from short-read sequences [14,15,16,17]. Nevertheless, the detection of SVs is an open question. The main challenge in detecting SVs by WES is due to the presence of repetitive sequences in the human genome and the location of breakpoints in genome regions that are troublesome for WES. Alignment errors, lack of sensitivity, complex SVs, high false-positive rates and the low signal from breakpoints within repetitive regions present other challenges [10,18,19]. WES also typically relies on paired-end reads 200–500 bp in size [20], whereas the average SV size in the human genome is 8 kb [21]. Moreover, detecting SVs from WES data is challenging as the evidence of SVs resembles standard sequencing and alignment artifacts [22,23]. Third-generation sequencing technologies developed by Pacific Biosciences [24] and Oxford Nanopore [25] overcome these challenges by generating long reads, even 100 kb or more in length. Nevertheless, despite longer reads, SVs larger than the average read size and those with breakpoints in repetitive regions remain challenging to detect. Additionally, these long-read sequencing methods are prone to a high per-base error rate and are costly [26]. Although methodological advances have enabled the detection of large SVs by sequencing and mapping of long reads, several technical and analytical limitations still hinder the successful detection of large SVs. Compared to WES, whole-genome sequencing (WGS) is capable of more accurate detection of copy number variations and produces less bias in identifying non-reference alleles [27]. However, WES is favored over WGS due to lower cost, faster turn-around-time, reduced data footprint and less complex data interpretation [28].

Similar to long-read approaches such as Single-Molecule Real-Time (SMRT) sequencing [24,25,26,29], Oxford Nanopore Technologies (ONT, Oxford, UK) [25] and synthetic long-read technology [30], linked-read sequencing [31] has been developed to improve SV detection across the genome. 10X Genomics (Pleasanton, CA, USA) developed the Chromium methodology, which provides linked-reads to reconstruct long DNA fragments by genome partitioning and barcoding [31]. Like the Pacific Biosciences (Menlo Park, CA, USA) and Oxford Nanopore technologies, the 10X Genomics linked-read method also uses a mapping-based approach where SVs are detected by mapping the reads directly to a given reference genome. The main advantage is the library preparation technique, which requires very low input (one nanogram) of high molecular weight DNA to generate sequence reads. It uses microfluidics partitions to produce droplet particles and avoids any fragmentation, making it ideal for sequencing high molecular weight DNA of 50 kbs or higher. More importantly, the linked-read method allows mapping the reads from the repetitive regions where SV breakpoints are often located [32,33]. Recent studies have demonstrated the utility of the linked-read method for detecting complex SVs in a variety of samples [34,35,36]. However, systematic evaluation of the performance of this relatively new method for detecting SVs is required. Moreover, its power in detecting SVs in hematological cancers, where high-quality DNA can easily be obtained, has not yet been assessed.

Previous studies comparing long read-based linked-read and short read-based sequencing technologies reported contradicting performances. Mark and colleagues compared linked-read sequencing with short read-based WGS as well as WES for SV detection [36]. The study reported that linked-read sequencing produced scalable and comprehensive information on SVs as compared to short-read sequencing. Later, Uguen et al. compared the linked-read method with short read-based WGS to detect SVs [33]. The study reported that linked-read sequencing could not improve the detection or characterization of SVs.

In this study, we assessed the performance of linked-read and WES methods for detecting SVs and short variants in samples from eight multiple myeloma (MM) patient cases. This cancer type provides an intriguing choice to systematically evaluate methods given that hypodiploidy and translocations are the predominant genetic alterations in MM and as MM tumors often encompass short variants like single nucleotide variants (SNVs) multinuclear variants and indels. Here, the performance of linked-read and WES to detect SVs was examined by comparing clinical cytogenetic data. The ability to detect short variants as well as MM-specific somatic mutations were assessed by integrative examination of linked-read, WES and RNA sequencing (RNA-seq) data, extending the scope of the study further to other types of genomic alterations.

## 2. Results

We evaluated the performance of linked-read sequencing and WES to detect SVs. We also compared the performance of linked-read sequencing, WES and RNA-seq to detect short variants. For these analyses, bone marrow (BM) aspirates with matched skin biopsies were collected from three MM patients and BM aspirates without matched skin biopsies from five additional MM cases. CD138+ plasma cells were enriched from the BM samples and DNA and RNA were extracted from the CD138+ cell fraction. Only DNA was also extracted from three skin samples. The clinical and cytogenetic characteristics of the patients are presented in Table 1. The samples were analyzed using linked-read, WES and RNA-seq methods. SVs were called using Manta SV caller, CNVkit and short variant using standard GATK variant calling pipeline, respectively (Figure 1).

### 2.1. Sequencing and Mapping Quality Statistics

The average total number of reads sequenced in the linked-read libraries (n = 11) was 77.74 million and mapped reads were 77.42 million with a 99.7% mapping rate. In WES libraries (n = 11), the average total number of reads was 129.12 million and the mapped reads were 129.02 million with a 99.9% mapping rate (Appendix A). The average mapped reads inside of regions were 57.72 million with 74.40% mapping rates in linked-read libraries. While in WES libraries, the average mapped reads inside of regions were 92.82 million with 73.31% mapping rates (Appendix A). However, the average on-target ratios were similar for linked-read and WES libraries, 74.62% and 73.38%, respectively (Appendix A).

The coverage was calculated for both linked-read (Figure 2a) and WES (Figure 2b), including matched skin (n = 3) and tumor samples (n = 8). The mean coverage inside of regions was 53 for linked-read and 139 for WES libraries, respectively (Appendix A). The genome fraction at different depts was calculated for both linked-read and WES. At the lower depths (<10×), both methods performed similarly. However, at the higher depths of 30× and 50×, WES had a higher percentage of bases captured (Figure 2c). The mean mapping quality inside of regions resulted in 57.82 and 58.28 for linked-read and WES, respectively (Figure 2d). Next, we calculated median insert size in base pairs (bp) and GC content for the linked-read and WES methods. The average insert size was 169 bp and 185 bp for linked-read and WES samples, respectively (Appendix A), while the average GC content was 49% for both linked-read and WES (Appendix A).

### 2.2. Detection of Total SVs by Linked-Read Sequencing and WES

To evaluate the efficiency of linked-read and WES for detecting total SVs, we compared SVs identified using the two methods applied to the same eight MM patient samples. The Manta SV caller identified five types of genomic alterations, including deletions, duplications, insertions, inversions and translocations. The analysis revealed discrepancies in the average SV counts produced by the two sequencing methods across the eight samples. Altogether, linked-read and WES reported an average total of 18,455 and 4065 SVs in the eight MM analyzed samples, respectively. The overlaps were calculated using bedtools with a minimum cutoff of 70% overlap. The two methods resulted in only 92 overlapping SVs. Of the total number of duplications identified by both methods, linked-read sequencing detected an average of 88.3% of the duplications (n = 1225) compared to WES, which detected only 9.8% (n = 111), with only 1.7% (n = 21) of the identified duplications overlapping between the two methods (Figure 3a). For the total number of inversions identified, linked-read sequencing detected over 99% (n = 17,026) of these SVs, whereas WES detected less than 1% (n = 49) (Figure 3b). In contrast, WES detected 95.7% (n = 2254) of the total deletions identified compared to linked-read sequencing, which detected an average of 2.6% (n = 56), where only 1.4% (n = 34) of all the identified deletions overlapped between the two methods (Figure 3c). In the case of translocations, WES detected 72.8% (n = 267) compared to linked-read sequencing, which detected 25.4% (n = 53), where only 1.8% (n = 4) of identified translocations overlapped (Figure 3d). For the total number of identified insertions, WES detected over 99% (n = 1323), whereas linked-read completely failed to detect insertions (Figure 3e). Overall, the distribution of SV counts from linked-read sequencing analysis indicated that 92.4% of the SVs were inversions, while other types of SVs were poorly detected. Overall, WES analysis detected all five types of SVs, including deletions (56.3%), insertions (32.6%), translocations (6.7%), duplications (3.3%) and inversions (1.2%) (Figure 3f), indicating that WES performs better for overall SV detection compared to linked-read sequencing, which mostly detected inversions. The absolute numbers are provided in Appendix A. Next, the analysis was restricted to high-quality SVs passing all the Manta quality filters, which reduced variant calls significantly but retained the trend of several fold higher amount of SV calls in linked-read datasets, linked-read: 67 duplications, 230 inversions and seven deletions where translocation and, insertions were not identified. In the case of WES: 19 duplications, nine inversions, 68 deletions, three insertions where no translocations were identified.

### 2.3. Detection of Somatic SVs by Linked-Read Sequencing and WES

Somatic SVs are genomic alterations that arise in tumors and can include genomic rearrangements, duplications or deletions of large DNA segments. Somatic SVs are critical for initiating and driving tumor development. For three of our samples, a matched skin sample was available and the cancer-specific somatic alterations specific in a given tumor sample (CD138 + cells) were detected by filtering SVs with those found in the skin DNA samples from the same patient. The results were in large part similar to those obtained with total SVs and revealed discrepancies in the average somatic SV counts produced by the two sequencing methods across the three sets of samples. Linked-read sequencing detected 2640, while WES detected only 50 somatic SVs in the same set of samples. Only seven somatic SVs overlapped between both methods. While linked-read sequencing detected a total of 625 somatic duplications, WES detected four, with only two duplications overlapping between both methods (Appendix A). Linked-read detected a total of 1985 inversions and WES detected only five inversions where only two overlapped (Appendix A). In contrast, WES detected 24 deletions and linked-read sequencing detected eight, where only three overlapped (Appendix A). WES detected 16 translocations and linked-read sequencing detected 12, where none of the translocations overlapped (Appendix A). Both methods failed to detect insertions in the somatic data. Overall, the distribution of SV counts detected by linked-read sequencing in the paired samples suggested that 75% of the SVs were inversions and 24% duplications, leaving other types of SVs poorly detected. Whereas the WES method detected 49% deletions, 32.7% translocations, 10% inversions, 8.2% duplications (Appendix A), suggesting that WES is a more efficient method for detecting somatic SVs. The absolute numbers of somatic SVs are provided in Appendix A. Next, the analysis was restricted to high-quality somatic SVs passing all the Manta quality filters, which reduced variant calls significantly but retained the trend of several fold higher amount of somatic SV calls in linked-read datasets, linked-read: 187 duplications, 397 inversions and five deletions where translocation and, insertions were not identified. In the case of WES: 14 deletions, four inversions and two duplications were identified.

### 2.4. Performance Evaluation for Detecting Known Clinical Cytogenetic Alterations

The genetic landscape of MM is complex and mainly includes hyperdiploidy defined as gains of chromosomes, and by chromosomal translocations involving the immunoglobulin heavy chain (*IGH*) gene. Cytogenetic events assessed by fluorescence in situ hybridization (FISH) is part of routine diagnosis and prognosis for MM patients in the clinic. We considered clinical cytogenetic alterations as the gold standard to evaluate linked-read and WES performance to detect clinically relevant SVs (Appendix A). We identified a total of ten different MM-associated cytogenetic alterations in our samples by FISH, including the translocations t(4;14), t(6;14), t(11;14), t(14;16) and t(14;20), plus recurrent chromosomal gains and losses, including gain(1q), del(1p), del(13q), del(14q) and del(17p). Next, we accessed the performance of linked-read sequencing and WES to detect these alterations. The true positive rate (sensitivity) is plotted against the false positive rate (100-specificity) to generate the receiver operating characteristic (ROC) curve and to calculate the area under the curve (AUC). Neither method could detect all of the cytogenetics events well in this test. The WES method, however, systematically surpassed the linked-read method and generated 3% better AUC measurements (Figure 4a,b).

Next, we analyzed copy number variations (CNVs) detected using linked-read and WES methods. The CNVkit [37] tool was used to detect and visualize the copy number variants. A flat reference was created using reference genome, target and anti-target interval files for each probe. Although the overall CNV data is comparable between linked-read and WES methods, several discrepancies were observed (Figure 5). We observed no distinct pattern of discrepancies specific to chromosomes or type of CNVs. Linked-read sequencing detected a total of 1373 copy number segments, with 843 duplications and 530 deletions. WES detected 1160 events with 658 duplications and 502 deletion events. In contrast to Manta, both methods detected more duplications than deletions. Next, the clinically relevant losses and gains were compared between linked-read and WES in the same samples del(13q) was only detected by WES in the MM_06_BM sample while del(17p) found in the MM_07_BM sample using both the methods. Although gain(1q) was detected by both methods in three samples, linked-read failed to detect this alteration in the MM_06_BM sample.

### 2.5. Performance Evaluation for Detecting MM-Specific SV Hotspots

To determine the competency of linked-read sequencing and WES at detecting myeloma-specific SVs, we used publicly available data from a recent large-scale study that comprehensively characterized SVs in 752 MM patients and identified 68 SV hotspots by low-coverage large-insert whole-genome sequencing [7]. Among the SV hotspots, 49 (>70%) have been identified as the gain of function hotspots and 19 (<30%) as loss of function hotspots. The gain of function hotspots included copy number gains, translocations, inversions and insertions. Loss of function hotspots included mostly deletions, also complex SVs and inversions. A median of two hotspots per patient was reported in the study. We analyzed the SVs detected by both linked-read sequencing and WES independently in our MM samples using these hotspots. Linked-read detected an average of 43.8 hotspots with a distribution of 71.8% (n = 31.5) gains, 22.8% (n = 10) losses and 5.4% (n = 2.38) fragile types in our samples. The median number of SV hotspots detected per sample by linked-read sequencing and WES were 41.5 and 3.5, respectively. The hotspots identified in each patient are available in Appendix A. The high number of hotspots detected by linked-read sequencing is due to the over-capture of inversions and duplications compared to WES. On average, WES detected 3.38 SVs with a distribution of 14.8% (0.5) gains, 51.9% (1.75) losses and 33.3% (1.13) fragile hotspots in our samples (Appendix A). The higher numbers of losses and fragile types are likely due to the high numbers of deletions detected by WES in our samples. However, the surprisingly large number of hotspots detected by linked-read sequencing suggests the possibility of a high false-positive rate by this method.

### 2.6. Detection of Total Short Variants by Linked-Read Sequencing and WES

We next evaluated the performance of linked-read sequencing and WES for detecting short variants. For both methods, short variants were called using the GATK best practice exome sequence analysis pipeline [38,39]. For the total number of short variants detected, linked-read sequencing detected 73% (n = 704) and WES detected 25% (n = 109) average short variants across the eight MM samples, with only 1.8% (n = 15) overlap (Figure 6a). Two samples, MM_04_BM and MM_05_BM, were outliers, where WES detected a higher number of short variants compared to the linked-read method. Next, we compared the number of somatic short variants in three paired samples. The linked-read method detected 94.3% (n = 673) and WES detected 4.9% (n = 37) average short variants, where less than 1% (n = 3) of the paired short variants overlapped (Figure 6b), suggesting that the linked-read sequencing detected a higher number of both total short variants and somatic short variants compared to WES. The absolute numbers of short variants are provided in Appendix A. The genes and overlaps are presented in Appendix A.

### 2.7. Performance Evaluation for Detecting Myeloma Specific Mutations

To determine the clinical applicability of the two methods, we assessed the concordance with specific somatic mutations that occur recurrently in MM. Variant calling was performed using the standard GATK pipeline for comparable results. For this comparison, MM-specific recurrent mutations were analyzed in five genes, including *BRAF*, *KRAS*, *NRAS*, *TP53* and *FAM46C* [40]. Linked-read sequencing detected seven out of a total of thirteen mutations detected by WES in the samples Figure 7a. Linked-read failed to detect *BRAF* (n = 2), *KRAS* (n = 2), *NRAS* (n = 1) and *FAM46C* (n = 2) in the samples. However, the failure to detect the mutations was not restricted to any particular type of mutation, e.g., frameshift, insertion or deletion. Further, we assessed gene-specific coverage and found that poor coverage and low quality in the linked-read data resulted in the discrepancies for detecting the mutations (Appendix A). For example, mutations to *NRAS* in MM_05_BM1 and *BRAF* in MM_04_BM were only detected by WES (Figure 7b,c). Next, we also assessed these mutations in the RNA-seq data derived from the same samples. By RNA-seq, we found nine out of thirteen mutations. However, RNA-seq analysis failed to detect mutations to *TP53*, *BRAF*, *KRAS* and *FAM46C* in three different samples.

### 2.8. Comparison of Short Variants and SVs Detected by Linked-Read Sequencing, WES and RNA-seq

To determine the overlap between technologies and to find out dataset-specific mutations, we compared linked-read sequencing, WES and RNA-seq derived short variants from the eight MM samples analyzed by all three methods (Appendix A). The maximum number of overlapping short variants among linked-read, WES and RNA-seq were 11 in patient MM_07_BM, while none of the overlapping short variants were found in patient MM_05_BM. Altogether, 25 overlapping short variants were detected by linked-read sequencing, WES and RNA-seq in the eight MM samples, while 113 overlapping short variants were identified by both linked-read sequencing and WES, 48 short variants by linked-read sequencing and RNA-seq, and 53 overlapping short variants by WES and RNA-seq. To compare SVs that could be detected by all three methods, we assessed if RNA-seq analysis could detect any of the fusion genes identified by linked-read sequencing or WES. However, none of the fusion genes identified by the DNA sequencing methods were detected by RNA-seq in any of the samples.

## 3. Discussion

Advances in next-generation sequencing technologies have enabled the investigation of the complexity of the human genome and the identification of structural variants in multiple cancer types [21,40,41]. Recurrent structural variations associated with different cancers signify the importance of studying these genomic alterations to better understand these diseases. Nevertheless, exome targeting sequencing technologies, which are still a popular technique, have had limited success in detecting SVs in genomic areas with high GC content, SV breakpoints in repetitive sequences, and large SVs. The 10X Genomics linked-read library preparation method has provided promising SV detection results by addressing challenges in exome sequencing. Although previous studies reported the ability of the linked-read technology to detect complex SVs [34,35,36], a comprehensive comparison of SV detection from hematological samples would help determine the added value of the more costly linked-read sequencing method in the diagnosis of these diseases from which high-molecular-weight DNA can easily be obtained. In this study, we compared the performance of whole-exome sequencing with the 10X linked-read methods for detecting SVs and short variants in samples from MM patients.

Our results suggest that WES performed better at detecting SVs compared to linked-read sequencing when applied to clinical samples. We used the Manta tool to call SV events independently from linked-reads and WES to produce comparable results. In our study, linked-read sequencing out-performed WES for detecting duplications and inversions but failed to detect insertions. Algorithms such as Long Ranger and LinkedSV have previously been used to detect SVs from linked-read sequence data; however, these tools are not designed to handle insertions [31,36]. Our findings are in line with previous reports inferring that linked-read sequencing can successfully detect deletion, duplication, inversion and translocation events except for insertions. Thus, the data analysis technique limits the detection of insertions using the linked-read method.

Linked-read sequencing was suggested to be capable of replacing commonly used assays for SV detection such as array comparative genomic hybridization and karyotyping [31,36], or to complement FISH [42], all of which are used in clinical diagnostics. However, our data indicated that linked-read sequencing was not able to detect common MM-related cytogenetic events that WES detected. In contrast to a previous study [43] where linked-read sequencing was able to detect disease-causal SVs missed by WES, we found that linked-read sequencing was unable to detect t(4;14) and del(13q) events in one sample, while WES did detect these alterations. Although both methods detected false-positive events, WES showed 3% better AUC measurements in identifying clinically relevant SVs. In the case of clinically relevant CNVs, neither CNVkit (Figure 5) nor Manta (Appendix A) results matched perfectly with the gold standard FISH data. For example, del(13q) was reported by FISH in four samples. However, Manta failed to detect this alteration in the MM_03_BM and CNVkit failed to detect it in the MM_01_BM sample. We were unable to provide evidence that linked-read sequencing could replace FISH in a diagnostic setting. In accordance with our results, a recent study also demonstrated that linked-read sequencing could not improve SV detection and characterization compared to short-read methods in a diagnostic setting [33].

Specialized visualization software designed for linked-read sequence data analysis such as Loupe have been used previously to detect such missed events [36]. Along with Loupe, complex events have been identified using advanced computational methods, special algorithms and other visualization tools specifically designed for linked-read sequencing such as Long Ranger and LinkedSV [31,35,36]. While significant computational advances have improved data visualization, it is equally essential to improve alignment and mapping techniques to increase data quality substantially, performance and sensitivity of the linked-read technology.

A landmark study of 752 patients established the SV landscape of multiple myeloma and reported a median of 2 SV hotspots observed per patient with this disease [7]. Our analysis compared MM-specific SV hotspots identified by Rustad et al. [7] using both linked-read sequencing and WES applied to a set of three germlines and tumor paired MM patient samples. The median of 3.5 SV hotspots detected by WES in our sample set was in line with the Rustad study. However, the median number of 41.5 SV hotspots detected by linked-read sequencing was comparatively high. We believe that the high number of SV hot spots detected in the linked-read data could be due to more false-positive events leading to discrepancies with the matched WES data. However, this finding should be interpreted with caution as the analysis was limited by a small sample size. Furthermore, WES reported more loss of function hotspots, whereas linked-read analysis reported more gain of function hotspots. These observations are in line with the higher number of deletions detected by WES and a higher number of duplications detected by linked-read sequencing.

Previous studies using primarily WES and array-based approaches have successfully contributed to identifying driver mutations in MM, including SNVs and copy number variations [4,44,45,46]. *NRAS*, *KRAS*, *BRAF*, *TP53*, *FAM46C*, *DIS3*, *CCND1*, and other genes, are thought to contribute to the pathogenesis of MM and are clinical biomarkers for the selection of targeted therapeutic strategies. Our study demonstrated that WES identified more somatic mutations than linked-read sequencing in five selected MM driver genes, including *NRAS*, *KRAS*, *TP53*, *FAM46C* and *BRAF*. In contrast, linked-read sequencing could detect just over 50% of the total somatic mutations detected by WES in these samples. These results contradict an earlier study indicating that linked-read sequencing facilitated the identification of mutations in disease-related genes otherwise difficult to sequence using WGS [36]. In addition, and unlike earlier studies, we were able to compare results between linked-read derived somatic mutation data with RNA-seq and WES data from the sample samples. Our results suggested both WES and RNA-seq were superior at detecting mutations in MM driver genes than linked-read sequencing analysis. We assume that the currently available short variant calling tools are not competent with linked-read sequencing. The tools might be inefficient at calling short variants from the long-reads in the case of linked-read WES as the long-read chemistry is designed to detect larger genomic alterations. Also, the lower coverage for linked-reads compared to WES leads to an inferior ability of the linked-read approach to detect driver mutations.

Our study focused on comparing data quality, SVs and short variant detection efficiency and clinically significant cytogenetic events and somatic mutation detection efficiency between WES and whole-exome linked-read methods. Our analysis suggests that WES outperformed linked-read sequencing to detect biologically and clinically relevant genomic alterations for our disease model of multiple myeloma. WES is considered a suboptimal method to detect SVs. Nevertheless, we found that WES detected the most clinically relevant SVs and short variants. It also was more cost-effective. Nevertheless, the linked-read approach could potentially help identify novel large SVs in different disease models with further development and improvement of downstream data analysis methods. Moreover, a comparison of WGS and linked-read whole-genome sequencing approaches could be more useful to evaluate the efficiency for investigating the panorama of genome-wide SVs. Taken together, we suggest that WES is a more powerful and robust approach for detecting cytogenetic alterations and somatic mutations in MM patients compared to linked-read sequencing. The same may also hold for other hematological cancers. Similar comparative evaluations between WES and linked-read sequencing with other state-of-the-art technologies specialized in detecting large SVs such as PacBio, Oxford Nanopore and Bionano Genomics technologies are warranted, especially to determine their applicability for diagnostic applications. Nevertheless, the low cost and established standard data analysis pipelines for WES make this an optimal method for current clinical implementation.

## 4. Materials and Methods

### 4.1. Patient Materials and Ethical Compliance

This study was conducted in accordance with the guidelines of the Declaration of Helsinki. BM aspirates (n = 8) and skin biopsies (n = 3) were obtained from MM patients after informed consent using protocols approved by an ethical committee of Helsinki University Hospital (study numbers 239/13/03/00/2010 and 303/13/03/01/2011). The data generated from the samples were stored in a secured server to protect data privacy and anonymity was maintained by encoding the sample identities.

### 4.2. Sample Processing

Bone marrow mononuclear cells (MNC) were isolated using Ficoll-Paque PREMIUM (GE Healthcare, Chicago, IL, USA) density gradient centrifugation. CD138+ cells were enriched from the MNC fraction using the EasySep™ positive selection kit from StemCell Technologies as described earlier [47]. DNA was isolated from CD138+ cells and skin biopsies using the DNeasy Blood and Tissue kit (Qiagen, Hilden, Germany). RNA was extracted from CD138+ cells using the miRNEasy kit (Qiagen, Hilden, Germany).

### 4.3. Whole-Exome Sequencing (WES)

The NEBNext^®^ DNA Library Prep Master Mix protocol (New England Biolabs, Ipswich, MA, USA) was used for the preprocessing of the DNA samples. Exomes were captured with the SeqCap EZ MedExome kit (Roche Nimblegen, Roche Nimblegen, Seattle, WA, USA), SureSelect Clinical Research Exome kit or the SureSelect Human All Exon V5 kit (Agilent Technologies, Santa Clara, CA, USA). The final libraries were sequenced using the HiSeq 2500 platform (Illumina, San Diego, CA, USA).

### 4.4. Linked-Read Exome Sequencing

Linked-read exome sequencing was used to sequence eight tumor samples and three skin samples. 1 ng of high molecular weight DNA was loaded onto a chromium controller chip. Linked-read libraries were processed according to the Chromium Exome Demonstrated Protocol (10x Genomics) with modifications in target enrichment (step 5.1–6.2). Hybridization was captured with the NimbleGen SeqCap EZ Library SR User’s Guide v5.1 protocols (Appendix A). 1000 ng per sample was pooled for five samples per capture. Blocking oligos were replaced with 10 µL of IDT xGen Universal Blockers—TS mix and 25 µL of COT. Samples were sequenced with the HiSeq 2500 instrument with paired-end 100 cycle runs using V4 chemistry. 6.4–9.9 gigabases were produced per sample. The data were analyzed as described previously [48].

### 4.5. RNA Sequencing

RNA-seq was performed as described earlier [49]. Briefly, mRNA was enriched from total RNA by depleting ribosomal RNA using the RiboZeroTM rRNA Removal kit (Illumina). A reverse transcription reaction was performed on the mRNA to synthesize complementary DNA (cDNA). From the cDNA, indexed libraries were constructed using the ScriptSeq V2™ Complete kit (Illumina), size selected and purified by agarose gel electrophoresis. Paired-end libraries were sequenced on the Illumina HiSeq 2500 instrument. RNA-seq data analysis was processed as described earlier [49] and included fusion gene calling with FusionCatcher [50] from RNA-seq fastq files and small variant analysis using GATK [51] best practice for transcriptome data.

### 4.6. Sequencing and Mapping Quality Statistics

Sequencing and mapping quality statistics for linked-read and WES were obtained with qualimap version 2.2 [52]. The quality metrics were combined using multiqc, version 0.8 [53]. Plots were generated using the multiQC and ggplot2 package in R.

### 4.7. Short Variant Calling

Small variants were called using the previously established GATK protocol [48]. Briefly, raw sequencing reads were trimmed and filtered using the Trimmomatic software [54]. Paired-end reads passing processing were then aligned to the GRCh38 human reference genome using Burrows-Wheeler Aligner, duplicates were marked with Picard, and alignment quality was improved using the Genome Analysis Toolkit [51] local realigner and base quality score recalibrator. Short somatic variants were then called using MuTect2 [55]. The analysis protocol with version information of all tools and details of reference datasets used in analyses have been explained earlier [48]. Following variant calling, variants were annotated with Annovar [56] and variants not passing all MuTect2 filters, falling into intronic and intergenic regions, classified as synonymous or non-frameshift variation, with an ExAC [57], ESP [58], 1KG [59] minor allele frequency higher than 1%, with a variant calling quality less than 40, residing in sites covered by less than 10 reads, with variant allele frequency less than 2% or higher than 30%, and with SNV strand-odd-ratio higher than 3 or indel strand-odd-ratio higher than 11 were removed.

### 4.8. Structural Variant (SV) Calling

Alignment files used in small variant discovery were also subjected to SV callings. SVs were called for eight tumor samples from both libraries using Manta [17] at default parameters. To collect the number of SVs detected in each sample the candidateSV.vcf.gz file generated by Manta tool was used. SVs and indel candidates also less than 50 bp in size from candidateSV.vcf.gz were used in Manta analyses. Given that filtering can exclude known variants and proper Manta filtering protocol is an open question, Manta variant calls were not filtered for quality. For the filtered analysis tumors.vcf.gz files were used. Somatic SVs were also called with Manta for three paired samples. In the case of somatic SVs candidateSV.vcf.gz files were used to count the number of detected SVs. For the filtered analysis somaticSV.vcf.gz were used.

### 4.9. Overlapping Variants

All overlapping SV and short variants were identified with bedtools version 2.29.1 [60] at 70% similarity at genomic positions. Candidate SVs identified by Manta and belonging to the same variation type in the WES and linked-read data were considered and overlapping variants were identified as unique WES features overlapping with linked-read features. Translocations were filtered before running bedtools due to the missing end position and compared separately in R. For paired samples, SV candidates passing the quality filters by Manta as somatic SVs were overlapped. Overlapping short variants were called similarly. Libraries Lattice, ggplot, dplyr, VennDiagram in R/Bioconductor software package (version 4.0) were used for visualization.

### 4.10. Somatic Mutation Overlap Analysis

To compare somatic mutations identified by linked-read sequencing, WES and RNA-seq, we focused on recurrently mutated genes in MM including *BRAF*, *KRAS*, *NRAS*, *TP53* and *FAM46C*. Somatic mutations were identified in the outputs of the GATK pipeline using linked-read sequence, WES and RNA-seq data.

### 4.11. Clinically Relevant Alterations

To identify clinically relevant SVs in our samples from the WES and linked-read sequence data, we relied on alterations identified by FISH, which is routinely performed at the clinic upon the patient’s diagnosis. Routinely checked events in MM including del(1p), gain(1q), del(13q), del(14q), del(17q), t(4;14), t(6;14), t(11;14), t(14;16) and t(14;20) were identified in our samples using deletion, duplication and translocation calls by Manta in the respective chromosome/chromosome arm. For translocations, the ID value was used to link breakend partners. SV candidates passing the quality filters by Manta as tumor SVs were used for this analysis.

### 4.12. Sensitivity and Specificity Calculation

The sensitivity and specificity were calculated considering cytogenetics events detected by FISH in the clinic as true labels. Altogether 14 cytogenetics events were detected across eight samples (Appendix A). The true positive rate (sensitivity) was plotted against the false positive rate (100-specificity) to generate the receiver operating characteristic (ROC) curve and to calculate the area under the curve (AUC) using the R package ROCR. The prediction and performance functions were used to calculate sensitivity and specificity. Every classifier evaluation using ROCR starts with creating a prediction object. The prediction function was used to transform the input data into a standardized format. Predictions; SVs detected using MANTA tool in WES and linked-read datasets independently. Labels; the true class labels considered from the FISH results (Appendix A). Next, predictor evaluations were performed by calculating sensitivity; P(Yhat = +|Y = +), estimated as: TP/P and specificity; P(Yhat = −|Y = −), estimated as: FN/P. Abbreviations; P (\# positive samples), N (\# negative samples), TP (\# true positives), TN (\# true negatives), FP (\# false positives), FN (\# false negatives).

### 4.13. Identification of SV Hotspots

To identify MM-specific SV hotspots, we relied on a recent study, which identified these hotspots using low-pass WGS [7]. The original publication used somatic SVs in paired samples, which were identified using Manta and Delly and passing the filters to identify the SV hotspots. The SV candidates passing the quality filters by Manta as tumor SVs were used to identify SV hotspots in our samples. Due to the limited number of paired samples, we used the tumor SV calls passing the filters from Manta in eight unpaired samples for this analysis. GRCh37 coordinates in the SV hotspot tables were converted to GRCh38 coordinates using the UCSC genome browser gateway. Two of the hotspots, involving chromosome 12 and chromosome X were not converted and therefore were excluded from further analysis. Using the position of the hotspots, SV events falling within these hotspots were identified. The number of hotspots was calculated for each patient. Lattice, ggplot, dplyr, Rcolorbrewer in the R/Bioconductor software package (version 4.0) were used for visualization.

### 4.14. Identification of Copy Number Variants (CNVs)

CNVkit [37] (version 0.9.8) was used for identifying copy number gains and losses in chromosomes in each tumor sample. Target and anti-target bed interval files were created according to the default pipeline using capture regions bed file for each probe. A flat reference was created with reference genome hg38 and respective target and anti-target interval files for each probe. Visualization was also performed with the CNVkit pipeline.

## 5. Conclusions

Our study systematically evaluated the performance and efficiency of WES and linked-read exome sequencing to detect SVs comprehensively and short variants highlighted discrepancies between methods, however, the outcome of these analyses demonstrated that WES out-performed linked-read sequencing for the identification of somatic and clinically relevant SVs and short variants. Although linked-read sequence analysis detected more events, WES identified more clinically relevant events and produced better coverage and mapping quality, indicating that WES is a more reliable method than linked-read sequencing for clinical implementation.

## Figures and Tables

**Figure 1 cancers-13-01212-f001:**
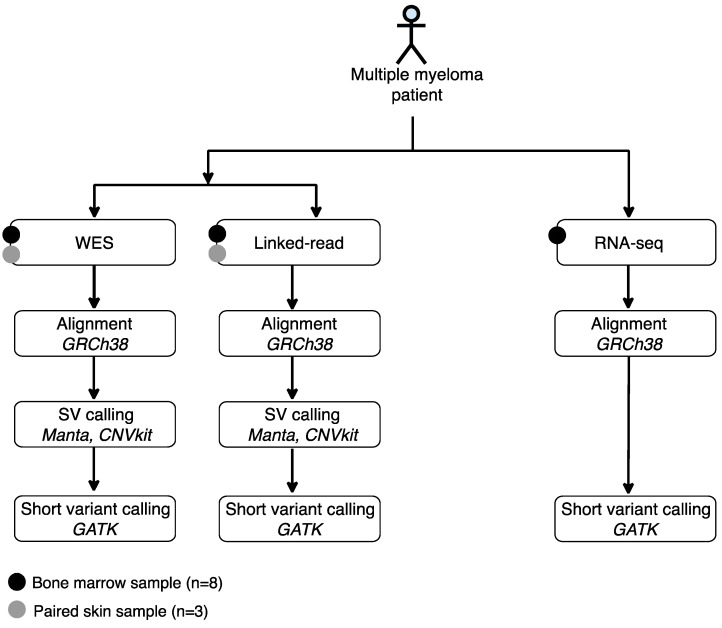
Experimental design and data analysis workflow of the study. Bone marrow aspirates were collected from eight multiple myeloma patient samples and three matched skin controls, followed by plasma cell (CD138+) isolation. Matched skin controls were collected from three patients. DNA and/or RNA were isolated and used for linked-read, whole-exome and/or RNA sequencing. The Manta SV caller and CNVkit were used to detect structural variants and the GATK variant calling pipeline was used to detect short variants.

**Figure 2 cancers-13-01212-f002:**
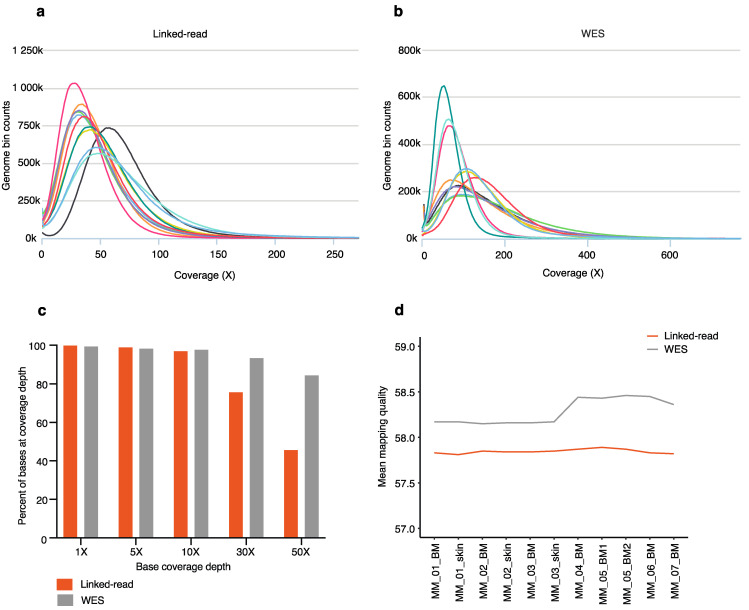
Sequencing and mapping quality statistics across all 8 MM samples and 3 skin samples. The coverage histograms for linked-read (**a**) and WES (**b**) show the number of reference bases plotted against the read depth (depth of coverage). The *x*-axis represents coverage, and the *y*-axis represents genome bin counts. Aggregating coverage values conveniently scale the bins of the *x*-axis. (**c**) The bar plot depicts the fraction of the genome with 1×, 5×, 10×, 30×, 50× coverage depths by linked-read and WES sequencing. The *x*-axis represents base coverage depth, and the *y*-axis represents the percent of bases at coverage depth. (**d**) The mean mapping quality inside of regions is plotted for each sample sequenced using both linked-read and WES methods.

**Figure 3 cancers-13-01212-f003:**
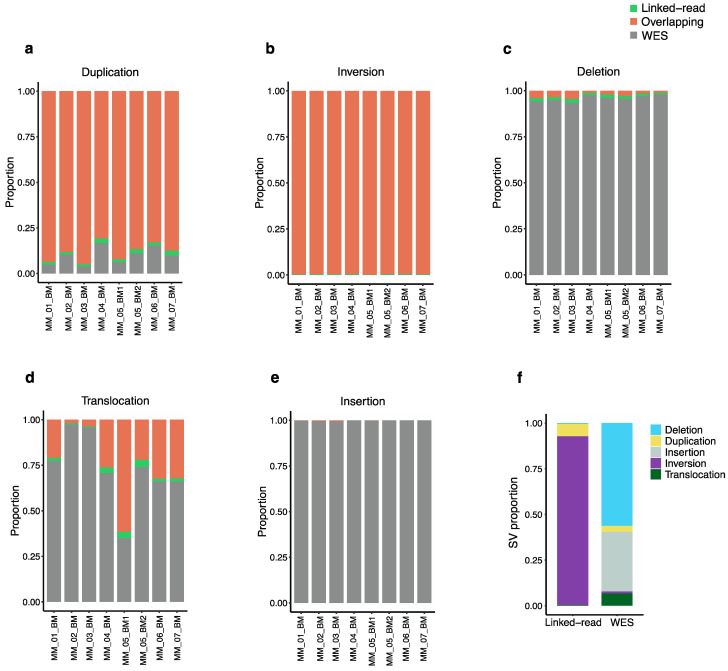
Total structural variants detected by linked-read sequencing and WES. The proportion of each type of structural variation separately across all 8 tumor samples. The orange bar represents the proportion of linked-read, the gray bar represents that of WES, and the green bar represents overlaps (**a**) Duplication; linked-read sequencing detected a higher proportion of duplication (**b**) Inversion; linked-read detected more than 99% of inversions called in each sample. (**c**) Deletions; WES detected more deletions across all the samples. (**d**) Translocations; with the exception of one sample, WES detected a higher proportion of translocations across the samples. (**e**) Insertions; WES detected 99.86% of all insertions. (**f**) The proportion of each type of SV is called by each sequencing method. Linked-read sequencing detected more duplications and inversions. No insertions were observed in the linked-read data as the insertion calls represent only 0.009% of total linked-read calls.

**Figure 4 cancers-13-01212-f004:**
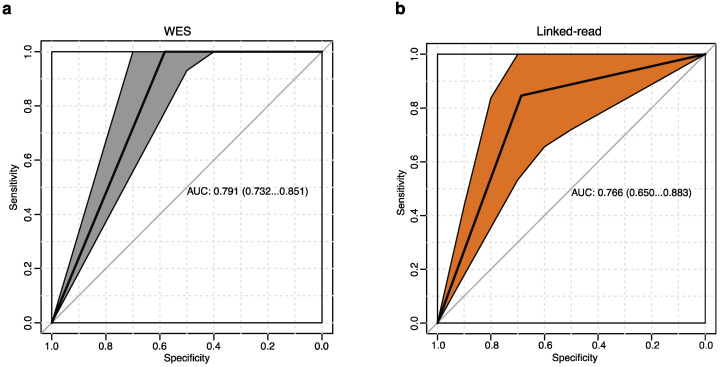
Receiver operating characteristics (ROC) analysis. (**a**,**b**) ROC curves of FISH-detected clinically relevant cytogenetics events compared with both linked-read and WES. The area under the curve (AUC) and confidence intervals are shown in the figures. The WES method provided higher AUC values. The *x*-axis represents specificity, and the *y*-axis represents sensitivity. (**a**) WES shows an AUC of 0.791 (**b**) Linked-read shows an AUC of 0.766.

**Figure 5 cancers-13-01212-f005:**
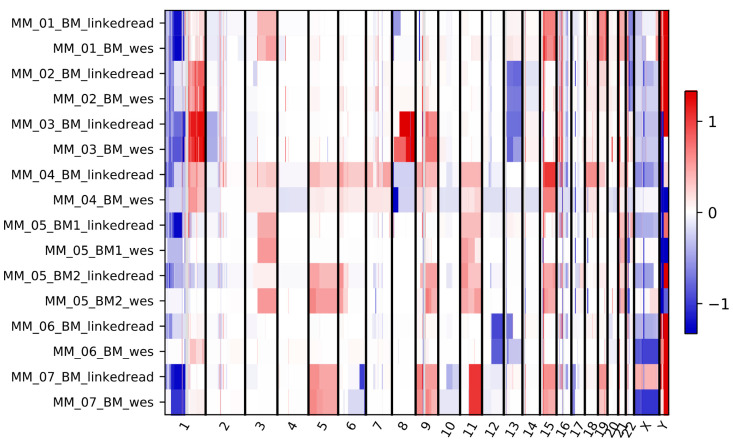
Copy number gain and loss profiles in each chromosome for eight tumor samples sequenced by both linked-read and WES methods. Red and blue represent gain and loss scores in the log2 scale, respectively. The scale −1 to 1 represent the log2 ratio value of a segment, weighted mean of all bins covered by the segment.

**Figure 6 cancers-13-01212-f006:**
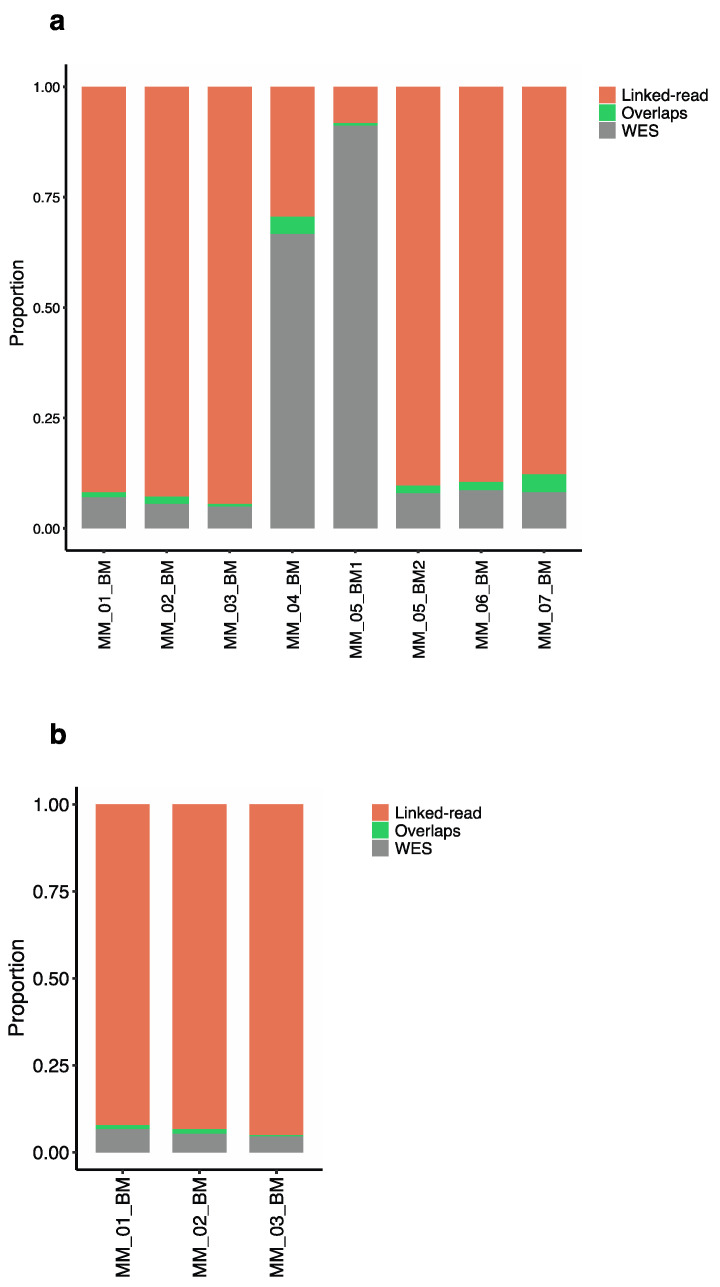
Short variants detected by the GATK pipeline in linked-read sequence and WES data. (**a**) The bar plot depicts the number of short variants detected by linked-read and WES methods in 8 tumor samples. Except for two samples, linked-read sequencing detected a higher proportion of short variants. (**b**) The short variants detected by GATK in 3 paired samples.

**Figure 7 cancers-13-01212-f007:**
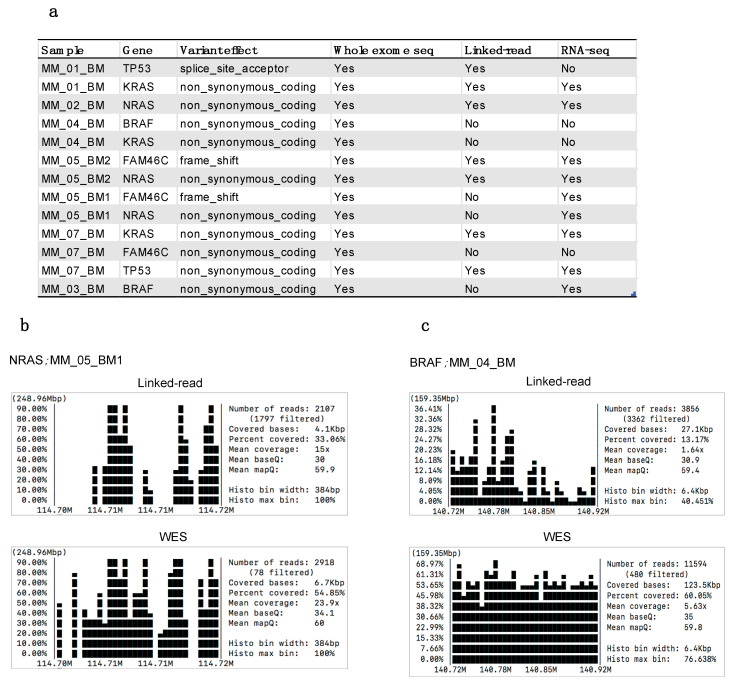
(**a**) List of myeloma-specific somatic mutations detected by whole-exome sequencing, linked-read and RNA-seq. (**b**,**c**) Gene coverage plots generated using SAMtools depicting *NRAS* in MM_05_BM1 and *BRAF* in MM_04_BM sample in linked-read and WES data.

**Table 1 cancers-13-01212-t001:** Clinical characteristics of the multiple myeloma patients.

Sample ID	Gender	Age at Diagnosis	Disease Status	Myeloma Characteristics	ISS Stage	WES	Linked-Read	RNA-Seq	Sample Type	Cytogenetics
MM_01_03	Male	57	Relapse	IgG lambda	3	Yes	Yes	Yes	Bone marrow and skin	del(13q), 1p loss
MM_02_03	Male	65	Relapse	IgG kappa	2	Yes	Yes	Yes	Bone marrow and skin	del (13q), possibly del(14)
MM_03_03	Male	60	Relapse	IgA lambda	3	Yes	Yes	Yes	Bone marrow and skin	del (13q), t(4;14), gain(1q), 14q32
MM_04_06	Female	69	Relapse	IgA, kappa	1	Yes	Yes	Yes	Bone marrow	gain (1q), trisomy 9, trisomy 11, Other deviation: trisomy 5, trisomy 15
MM_05_03	Male	56	Relapse	Unknown, kappa	1	Yes	Yes	Yes	Bone marrow	trisomy 9, trisomy 11, other deviation: trisomy 5, trisomy 15
MM_05_09	Male	56	Relapse	Unknown, kappa	1	Yes	Yes	Yes	Bone marrow	trisomy 9, trisomy 11, other deviation: trisomy 5, trisomy 15
MM_06_03	Male	41	Refractory	IgG lambda	ND	Yes	Yes	Yes	Bone marrow	Monosomy 13, del(13q)
MM_07_03	Male	56	Relapse	IgG kappa	ND	Yes	Yes	Yes	Bone marrow	del (17p), gain (1q), 1p36 loss

## Data Availability

The data presented in this study are available on request from the corresponding author. The data are not publicly available due to privacy and ethical limitations.

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
