# Peer review of "Comparison of Structural and Short Variants Detected by Linked-Read and Whole-Exome Sequencing in Multiple Myeloma"

_cancers, 2021, doi:10.3390/cancers13061212_

Round 1
Reviewer 1 Report
Dear Authors,
Your systematic study is presented in a very interesting overview including clear figures and supplementary data. However, a few comments in the analysis and quality of this data;
In paragraph 4.3 in the material and methods part, it seems that two different types of WES enrichment are used in the sample preparation for the exome sequencing is this correct including which samples? Both WES bed file links should be included in the manuscript to observe the content of the exome genes and parts. Further, is the comparison between the Nimblegen/Roche MedExome kit for the 10X Genomics linked-reads versus the standard Agilent SureSelect Exome V5 version the results due to the content of probes? And is the content (Mb) of the probes in the enriched driver genes similar in both library preparations? No missing reads in the NGS data for the selected driver genes?
In paragraphs 4.3 and 4.4 in the material and methods part, I miss the quality scores of the sequencing reads and values of the trimming reads in the analysis. I'm asking this since if the NGS data output is probably different for both paragraphs, it's difficult to compare the data output, and otherwise, it seems like comparing apples and pears since your conclusion is very strong in this study about 10X Genomics method. Further, indicating the missing somatic variants in the driver genes in the linked-read WES will also show a high level of missing variants in a complete linked-read WES data versus WES and RNA sequencing data. Could you comment on this why this is happening in these two approaches in your manuscript? And just for my interests, did you report these results back to 10X Genomics?
Kind regards, reviewer
Author Response
Rebuttal letter
Reviewer 1
Your systematic study is presented in a very interesting overview including clear figures and supplementary data.
We appreciate the efforts of the reviewer in thoroughly going through the manuscript and providing constructive comments.
However, a few comments in the analysis and quality of this data; In paragraph 4.3 in the material and methods part, it seems that two different types of WES enrichment are used in the sample preparation for the exome sequencing is this correct including which samples?
We agree with the reviewer that using different kits for WES enrichment is not ideal for the comparative analysis. However, WES was routinely performed as part of the prospective real-time molecular profiling for hematology patients in a precision medicine setting to guide treatment decisions. Due to technological advancements, the kits were intermittently changed to produce better quality data. Importantly, our data analysis pipelines and quality control methods were monitored to ensure the detection of clinically relevant variants in an individual patient sample. We have added the kit information for each sample in Supplementary Table 1 in the modified manuscript.
Both WES bed file links should be included in the manuscript to observe the content of the exome genes and parts.
We thank the reviewer for the suggestion. We have included the bed files in the supplementary material in the modified manuscript version.
Further, is the comparison between the Nimblegen/Roche MedExome kit for the 10X Genomics linked-reads versus the standard Agilent SureSelect Exome V5 version the results due to the content of probes? And is the content (Mb) of the probes in the enriched driver genes similar in both library preparations? No missing reads in the NGS data for the selected driver genes?
We agree with the reviewer’s concern. All the sequencing experiments were performed by a core-service laboratory at the FIMM Technology Center. The Technology Center experts test and compare different kits to assess reproducibility and robustness before making a decision to change the kit for routine exome-sequencing experiments. Therefore, we believe that the content of probes and other confounding factors should have minimum interference in detecting variants, especially biologically/clinically relevant variants. We have now added the kit information for each sample in Supplementary Table 1.
In paragraphs 4.3 and 4.4 in the material and methods part, I miss the quality scores of the sequencing reads and values of the trimming reads in the analysis. I'm asking this since if the NGS data output is probably different for both paragraphs, it's difficult to compare the data output, and otherwise, it seems like comparing apples and pears since your conclusion is very strong in this study about 10X Genomics method.
We agree with the reviewer’s comment. We have added the quality matrix, for example, mean mapping quality, a total number of reads, mapped reads, coverage at different cutoffs for all the samples in Supplementary Table 1 in the modified manuscript. Also, we have added a new plot in Figure 2b depicting the mean mapping quality calculated using the qualimap tool. We have also modified the text accordingly (line number 204-207).
Further, indicating the missing somatic variants in the driver genes in the linked-read WES will also show a high level of missing variants in a complete linked-read WES data versus WES and RNA sequencing data. Could you comment on this why this is happening in these two approaches in your manuscript? And just for my interests, did you report these results back to 10X Genomics?
We can only speculate why the linked-read WES method detected a smaller number of overall variants and variants in driver genes compared to WES and RNA-sequencing. We assume that the variant calling tool is not competent with this emerging technology. The tool might be inefficient in calling variants from the long-reads in the case of linked-read WES. We think that incompetencies at experimental and data analysis levels had contributed to the inefficiency of the linked-read method. We have added this sentence in the discussion in the modified manuscript (line number: 591-606)
We agree with your suggestion to report the results to 10X Genomics. Nevertheless, 10X Genomics has discontinued providing the linked-read exome-sequencing technology due to apparent shortcomings.

Reviewer 2 Report
Kumar et al. present in their manuscript „Comparison of structural and short variants detected by linked-read and whole-exome sequencing in multiple myeloma” a comprehensive study comparing the ability of linked-read WES and WES analysis to identify structural variations (SV) in tumor samples. They showed that SV called from linked-read WES and WES datasets exhibit only a weak overlap. WES analysis seems to be superior in detection of clinical relevant SV and short variants. Therefor they expect no benefit from using linked-read WES in routine diagnostic setup.
Major concerns
The autors should include at least one GIAB sample to their analysis. This would allow them to estimate the false-positive and false-negative rates of the discovered SV. Without it is hard to conclude which of the both methods is better to identify SV. Although they verify some of the multiple myeloma-associated SV with FISH, this is only a very tiny amount compare with all the SV are identified and therefore hardly representative.
A CNV analysis (e.g. CNVkit) should be include in the bioinformatics pipeline for a comprehensive evaluation of deletions and amplification. A deletion of a whole chromosome arm (like the descript del(13q)) should be easily detectable in linked-read WES and WES datasets.
Commonly used coverage statistics for WES should be included (see comment of Fig.2)
In chapter 2.7 the authors claimed that linked-read WES miss more short variants than the WES analysis. It would be helpful to show the mean, min and max coverage of the genes used in this comparison. I would doubt that linked-read WES would miss all this variants at the same coverage.
Minor concerns
Perhaps I missed this, but statistics regarding the overlap of SV and short variants in the corresponding WES BM and WES skin samples would be helpful.
A short comparison of linked-read WES/WES with state-of-the-art technologies to detect SV in cancer (PacBio, Oxford Nanopore Technologies and Bionano) in the discussion would be beneficial.
In chapter 2.8 the authors descript that they were not able to find the potential fusions genes identified by linked-read WES and WES in the RNA-seq analysis. Therefore it would good to know, if the transcripts of the potential fusions genes where expressed and thus detectable. If yes, this could be hint, that a certain amount of the discovered SV could be artefacts.
Fig2: The Authors showed a whole genome coverage plot, which is perhaps not the best way for WES analysis. A table including on-target rate, percentage of ref seq exons covered at 10x, 20x and 30x, GC% and such parameters would be much more helpful.
Changing the labeling to MM_01_BM and MM_01_skin would be easier to read instead of MM_01_01 and MM_01_03.
In the figure legend (line 158) the bp is missing at the insert size.
Line 161: The comparison of the GC content should be dismissed. Due to the different WES enrichment chemistries they use, it is hard to estimate if this small difference is due to the linked-read WES and WES approached or due to the different chemistries, which clearly have a substantial impact.
Line 275: MM_04_06 and MM_05_03 are descript showing a higher number of short variants. Therefore, it would be essential to know, which sample was prepared with which enrichment chemistry, to exclude effects of the library preparation.
Line 336: The authors claimed that WES showing better mapping quality and coverage. This should proofed by statistical analysis.
Author Response
Rebuttal letter
Reviewer 2
Kumar et al. present in their manuscript „Comparison of structural and short variants detected by linked-read and whole-exome sequencing in multiple myeloma” a comprehensive study comparing the ability of linked-read WES and WES analysis to identify structural variations (SV) in tumor samples. They showed that SV called from linked-read WES and WES datasets exhibit only a weak overlap. WES analysis seems to be superior in detection of clinical relevant SV and short variants. Therefor they expect no benefit from using linked-read WES in routine diagnostic setup.
We appreciate the efforts of the reviewer in thoroughly going through the manuscript and providing constructive comments.
Major concerns
The autors should include at least one GIAB sample to their analysis. This would allow them to estimate the false-positive and false-negative rates of the discovered SV. Without it is hard to conclude which of the both methods is better to identify SV. Although they verify some of the multiple myeloma-associated SV with FISH, this is only a very tiny amount compare with all the SV are identified and therefore hardly representative.
We agree with the reviewer that a “genome in a bottle (GIAB)” control sample could have been very useful to estimate false-positive and false-negative rates to compare the quality assessments between linked-read WES and WES techniques. However, adding the GIAB control would require that all the experiments are performed again and it would be challenging to report the results in a short period of time. Also, as per our understanding, GIAB would serve as a good control only for short variants but not for SVs.
A CNV analysis (e.g. CNVkit) should be include in the bioinformatics pipeline for a comprehensive evaluation of deletions and amplification. A deletion of a whole chromosome arm (like the descript del(13q)) should be easily detectable in linked-read WES and WES datasets.
We appreciate the constructive input from the reviewer. We have now added results from the CNVkit tool with the results summarized in the revised manuscript in Figure 5. The result section describing CNV alternations is described from (line number 326-340). In line with the reviewer’s assumption, we clearly observed 13q deletion and other copy number variations. The analysis has indeed improved the manuscript. We have also mentioned CNVkit in the discussion (line number: 545-552). The materials and method section has also updated (line number: 793-800)
Commonly used coverage statistics for WES should be included (see comment of Fig.2)
We agree with the reviewer’s comment. We have now added the quality matrix e.g. mean mapping quality, a total number of reads, mapped reads, coverage at different cutoffs for all the samples in Supplementary Table 1. Also, we have added a new Figure 2b depicting the mean mapping quality calculated using the QualiMap tool. We have also modified the text accordingly in section 2.1.
In chapter 2.7 the authors claimed that linked-read WES miss more short variants than the WES analysis. It would be helpful to show the mean, min and max coverage of the genes used in this comparison. I would doubt that linked-read WES would miss all this variants at the same coverage.
To address this comment we generated gene-specific coverage plots and added them to Figure 7b-c and Supplementary Figure 4 in the revised manuscript (line number: 459-463).
Minor concerns
Perhaps I missed this, but statistics regarding the overlap of SV and short variants in the corresponding WES BM and WES skin samples would be helpful.
We added the data on overlapping SVs and short variants between paired skin and MM samples in Supplementary Table 3 (Number of SVs detected in tumor, skin and overlaps). The number of short variants detected in tumor, skin and overlaps is added to Supplementary Table 7.
A short comparison of linked-read WES/WES with state-of-the-art technologies to detect SV in cancer (PacBio, Oxford Nanopore Technologies and Bionano) in the discussion would be beneficial.
The suggested comment will give a broader perspective on the latest technologies to detect SVs in the cancer and their limitations to the audience. We have now updated the discussion in the revised manuscript to orient linked-read WES methods with other existing state-of-art technologies for efficient SV detection in cancer (line number: 621-627).
In chapter 2.8 the authors descript that they were not able to find the potential fusions genes identified by linked-read WES and WES in the RNA-seq analysis. Therefore it would good to know, if the transcripts of the potential fusions genes where expressed and thus detectable. If yes, this could be hint, that a certain amount of the discovered SV could be artefacts.
We believe RNA-sequencing is a well-established method to detect expressed fusion genes. Considering RNA-seq data as a standard where not a single fusion was detected, we did not expect any fusions to be detected by linked-read or WES methods.
Fig2: The Authors showed a whole genome coverage plot, which is perhaps not the best way for WES analysis. A table including on-target rate, percentage of ref seq exons covered at 10x, 20x and 30x, GC% and such parameters would be much more helpful.
We agree with the reviewer and updated Figure 2a in the revised manuscript that includes coverage at 10X, 20X, 30X, and 50X. The data used to make the figure is available in Supplementary Table 1.
Changing the labeling to MM_01_BM and MM_01_skin would be easier to read instead of MM_01_01 and MM_01_03.
We agree that the suggested patient identifier indeed reads better. We have now changed the patient identifiers in the modified manuscript to MM_01_BM and MM_01_skin.
In the figure legend (line 158) the bp is missing at the insert size.
We thank the reviewer for identifying this mistake and have corrected the figure legend.
Line 161: The comparison of the GC content should be dismissed. Due to the different WES enrichment chemistries they use, it is hard to estimate if this small difference is due to the linked-read WES and WES approached or due to the different chemistries, which clearly have a substantial impact.
We think that the reviewer has a valid point. Considering this comment, we have now removed the results comparing GC content.
Line 275: MM_04_06 and MM_05_03 are descript showing a higher number of short variants. Therefore, it would be essential to know, which sample was prepared with which enrichment chemistry, to exclude effects of the library preparation.
We support the reviewer’s argument. The library preparation method for each MM patient sample is now shown in Supplementary Table 1 in the revised manuscript.
Line 336: The authors claimed that WES showing better mapping quality and coverage. This should proofed by statistical analysis.
We agree with the reviewer. Since the objective was to mention the overall trend of mapping quality and coverage, we have removed this sentence in the revised manuscript.

Reviewer 3 Report
In section 2.1, “The genome fraction at 10X depth was calculated for both linked-read and WES, including matched skin (n=3) and tumor samples (n=8). The analysis resulted in an average horizontal coverage of 3.31% for linked-read and 3.54% at a minimum depth of 10X for WES, respectively (Figure 2a).”
For WES and linked-read exome sequencing data, measuring the average horizontal coverage at the whole genome scale is irrelevant. What are the on-target ratios? What are the average depths (and the % coverage at a minimum depth of 10X) in exome regions?
The author should provide a table to list the basic sequencing stats for each sample, such as number of raw reads, the alignment ratios, on-target ratios, etc.
In section 2.2, the authors evaluate the efficiency of linked-read and WES for detecting total SV. The two methods resulted in only 92 SVs (this is less than 0.5% of the detected SVs!) overlapping with a minimum cutoff of 70% overlap.
If most of the detected SVs are real, it means the sensitivity of SV detection (i.e. duplications and inversions) based on the WES data is extremely low. Without a proper validation or a gold standard set, it is impossible to judge the efficiency and determine which method is better.
Again, in section 2.3, linked-read sequencing detected 2640 somatic SVs while WES detected only 50 somatic SVs in the same set of samples. Only 7 somatic SVs overlapped between both methods. Without a proper validation or a gold standard set, this comparison is not useful.
In the AUC plots (Figure 4 a,b), it seems that the WES data can detect all of the cytogenetics events (i.e. Sensitivity = 1) with the 60% specificity. However, the authors claimed that neither method could detect all of the cytogenetics events well in this test. How many cytogenetics events could be detected in each dataset? How did the authors calculate the sensitivity and specificity?
Lines 273-275: What caused these two outliers? Is it due to the sequencing depths or capture efficiencies?
Section 2.7, “…Variant calling was performed using the standard GATK pipeline for comparable results…”
The authors didn’t clearly describe how they filtered the variants from the standard GATK pipeline. I cannot find the description in the M&M.
In addition, it is not clear how many mutations were detected from each dataset. I expect there should be dataset specific mutations. However, the authors only mentioned the 13 mutations identified in the WES data, which makes the comparison biased.
In sections 4.3 & 4.7, the methods of variant calling and filtering are sparsely described. The authors simply referenced to ref #49 as the “GATK best practice exome-seq pipeline”, but this might not be appropriate.
For examples, it is not clear if the authors filtered variants against a panel of normals generated from the exome data of 24 healthy unrelated Finnish individuals, as described in ref #49.
Did somatic variants with coverage <=10 were filtered out, as also described in ref #49?
Did the authors use Haplotypecaller?
Author Response
Rebuttal letter
Reviewer 3
We appreciate the efforts of the reviewer thoroughly going through the manuscript and providing constructive comments.
In section 2.1, “The genome fraction at 10X depth was calculated for both linked-read and WES, including matched skin (n=3) and tumor samples (n=8). The analysis resulted in an average horizontal coverage of 3.31% for linked-read and 3.54% at a minimum depth of 10X for WES, respectively (Figure 2a).”
For WES and linked-read exome sequencing data, measuring the average horizontal coverage at the whole genome scale is irrelevant. What are the on-target ratios? What are the average depths (and the % coverage at a minimum depth of 10X) in exome regions? The author should provide a table to list the basic sequencing stats for each sample, such as number of raw reads, the alignment ratios, on-target ratios, etc.
We agree with the reviewer’s comment. We have now added the quality matrix e.g. mean mapping quality, a total number of reads, mapped reads, coverage at different cutoffs for all the samples in Supplementary Table 1. Also, we have added a new Figure 2b depicting the mean mapping quality calculated using the QualiMap tool. We have also modified the text accordingly (line number: 197-200; 204-207).
In section 2.2, the authors evaluate the efficiency of linked-read and WES for detecting total SV. The two methods resulted in only 92 SVs (this is less than 0.5% of the detected SVs!) overlapping with a minimum cutoff of 70% overlap.
If most of the detected SVs are real, it means the sensitivity of SV detection (i.e. duplications and inversions) based on the WES data is extremely low. Without a proper validation or a gold standard set, it is impossible to judge the efficiency and determine which method is better. Again, in section 2.3, linked-read sequencing detected 2640 somatic SVs while WES detected only 50 somatic SVs in the same set of samples. Only 7 somatic SVs overlapped between both methods. Without a proper validation or a gold standard set, this comparison is not useful.
We agree with the reviewer´s comment that a very small number of overlapping SVs and short variants between linked-read and WES is an unexpected finding. However, it is well established that WES is not an ideal method to detect SVs that is justified by a low number of SVs detected using WES compared to linked-read.
The best approach could have been to sequence the same samples with whole-genome sequencing to get a better perspective of SV and short variant detection. However, we did not perform the whole genome sequencing due to the cost limitations and scope of this study.
As it could be difficult to validate all the SVs and short variants, we validated clinically relevant cytogenetics events using the clinical FISH data as a gold standard.
In the AUC plots (Figure 4 a,b), it seems that the WES data can detect all of the cytogenetics events (i.e. Sensitivity = 1) with the 60% specificity. However, the authors claimed that neither method could detect all of the cytogenetics events well in this test. How many cytogenetics events could be detected in each dataset? How did the authors calculate the sensitivity and specificity?
We apologize for not providing this information in the first version of the manuscript. Now we have modified the text in the materials and methods section by adding a new section 4.12 “Sensitivity and specificity calculations” (line numbers: 760-774).
Lines 273-275: What caused these two outliers? Is it due to the sequencing depths or capture efficiencies?
To make sure why we have two outliers we reanalyzed the data and considered the following factors:
Library preparation kit; all the linked-read samples were prepared using Roche Nimblegen MedExome kit and WES libraries were prepared using either Roche Nimblegen MedExome (n=6) or Agilent SureSelect Clinical Research Exome (n=5).
Linked-read MM_04_BM, kit; Roche Nimblegen MedExome, Mapping quality; 42.273 (Linked-read average mapping quality across 11 samples = 42.02)
WES MM_04_BM, kit; Agilent SureSelect Clinical Research Exome, Mapping quality; 44.7268 (average mapping quality across 11 samples = 44.05)
Linked-read MM_05_BM1, kit; Roche Nimblegen MedExome, Mapping quality; 42.168 (linked-read average mapping quality across 11 samples = 42.02)
WES MM_05_BM1, kit; Agilent SureSelect Clinical Research Exome, Mapping quality; 44.7612 (WES average mapping quality across 11 samples = 44.05)
Based on other read statistics both samples are in the middle range. We can speculate that biology might have a role to play, however, we can not make a statement regarding the confounding factors for the variation in these particular samples.
Section 2.7, “…Variant calling was performed using the standard GATK pipeline for comparable results…”
The authors didn’t clearly describe how they filtered the variants from the standard GATK pipeline. I cannot find the description in the M&M.
We apologize for not providing this information in the first version of the manuscript. This information has now been included in the manuscript chapter 4.7.
In addition, it is not clear how many mutations were detected from each dataset. I expect there should be dataset specific mutations. However, the authors only mentioned the 13 mutations identified in the WES data, which makes the comparison biased.
We agree with the reviewer and elaborated dataset-specific mutations in the modified manuscript in section 2.8. The results are summarised from line number: 480-492 and in Supplementary Figure 3 a-h. The figures illustrate the number of mutations detected in each dataset as well as the overlapping mutations. The objective was to first compare overall detected mutations and then focus on myeloma-specific short variants.
In sections 4.3 & 4.7, the methods of variant calling and filtering are sparsely described. The authors simply referenced to ref #49 as the “GATK best practice exome-seq pipeline”, but this might not be appropriate.
For examples, it is not clear if the authors filtered variants against a panel of normals generated from the exome data of 24 healthy unrelated Finnish individuals, as described in ref #49.
Did somatic variants with coverage <=10 were filtered out, as also described in ref #49?
Did the authors use Haplotypecaller?
We apologize for not providing this information in the first version of the manuscript. A more in-depth description of short variant discovery has now been included in the manuscript chapter 4.7 (line number: 658-699).
Briefly, our approach relied on the protocol listed in ref Dufva et al, 2018 with some modifications in variant filtering. The approach also included filtering of variants against the same normal-panel given that normal-panel samples, samples of this study, and samples of study of Dufva et al, 2018 were all prepared and sequenced at the same sequencing laboratory using more or less the same instruments and protocols.
Somatic variants residing in genome regions with total coverage of less than 10 reads were filtered. This is now explained in the manuscript.
Variants in this and ref49 were called using MuTect2. Postprocessing included estimation of cross-contamination and sequencing artifacts using GATK tools as detailed in ref 49.

Round 2
Reviewer 2 Report
The authors greatly improved the manuscript! Therefor I would strongly recommend to accept and publish the manuscript, with 3 minor changes and without further revision.
1)line 259: Full stop is missing.
2)line 474-477: The lower coverage for linked-reads vs WES should be mentioned as cause for inferior ability of linked-read approach to detect driver mutations.
3)line 495: Regarding the homepage, Oxford Nanopore is written with a minor p.
Author Response
Review Report (Reviewer 2) is attached.

Reviewer 3 Report
What does “Std coverage data” mean in Supplementary Table 1? The mean coverage seems very low for all of the samples (column L in Supplementary Table 1). What are the on-target ratios?
I am concerned that the power of detecting SVs using such low coverage data is poor, which may result to the following finding - a very small number of overlapping SVs and short variants between linked-read and WES.
Author Response
The review report is attached.
